# Phenolic Profile, Antioxidant Activity, and Chemometric Classification of Carob Pulp and Products

**DOI:** 10.3390/molecules28052269

**Published:** 2023-02-28

**Authors:** Georgia D. Ioannou, Ioanna K. Savva, Atalanti Christou, Ioannis J. Stavrou, Constantina P. Kapnissi-Christodoulou

**Affiliations:** 1Department of Chemistry, University of Cyprus, 1678 Nicosia, Cyprus; 2Department of Life Sciences, European University Cyprus, 2404 Nicosia, Cyprus

**Keywords:** carobs, carob-derived products, polyphenols, antioxidant activity, HPLC, chemometrics, PCA, OPLS-DA

## Abstract

In recent years, carob and its derived products have gained wide attention due to their health-promoting effects, which are mainly attributed to their phenolic compounds. Carob samples (carob pulps, powders, and syrups) were analyzed to investigate their phenolic profile using high-performance liquid chromatography (HPLC), with gallic acid and rutin being the most abundant compounds. Moreover, the antioxidant capacity and total phenolic content of the samples were estimated through DPPH (IC_50_ 98.83–488.47 mg extract/mL), FRAP (48.58–144.32 μmol TE/g product), and Folin–Ciocalteu (7.20–23.18 mg GAE/g product) spectrophotometric assays. The effect of thermal treatment and geographical origin of carobs and carob-derived products on their phenolic composition was assessed. Both factors significantly affect the concentrations of secondary metabolites and, therefore, samples’ antioxidant activity (*p*-value < 10^−7^). The obtained results (antioxidant activity and phenolic profile) were evaluated via chemometrics, through a preliminary principal component analysis (PCA) and orthogonal partial least square-discriminant analysis (OPLS-DA). The OPLS-DA model performed satisfactorily, differentiating all samples according to their matrix. Our results indicate that polyphenols and antioxidant capacity can be chemical markers for the classification of carob and its derived products.

## 1. Introduction

The carob tree (*Ceratonia siliqua* L.), which belongs to the Leguminosae family, is a flowering evergreen shrub widely cultivated in the Mediterranean region and the Middle East. The carob or carob pod is the edible fruit produced from the carob tree and has recently attracted great attention due to its unique chemical composition and health-promoting properties. The carob pod consists of two main parts, the pulp and the seeds in a ratio of about 90:10 (*w*/*w*) [1,2]. Even though carob seeds represent only a small part of the total weight of the fruit, they are of great industrial interest, as they are used to produce locust bean gum (LBG, E410), a natural food thickening and stabilizing agent [1,3]. The pulp remaining after the industrial exploitation of the seeds, also referred to as carob kibble, is considered an agri-food waste [4,5]. Carob pulp has, so far, been used as livestock feed, while recently the worldwide market has turned its attention to carob pulp as it is considered a promising functional and nutraceutical food component [3,6,7]. Currently, carob pulp is used as a raw material for the preparation of several organic products, such as roasted and unroasted carob powders (a cocoa substitute, free of caffeine and theobromine) and carob syrups or molasses [6].

Carob has recently attracted great attention due to its health-promoting effects and potential use for value-added food production. It is considered an abundant source of several beneficial substances, including dietary fibers, carbohydrates, minerals, and polyphenolic compounds [8]. The majority of the biological effects and health benefits of carob have been associated with the presence of polyphenols [9]. Phenolic compounds are a widespread class of compounds, produced by the secondary metabolism of plants and present in all species of plant origin. Polyphenols are divided into the following subcategories: phenolic acids, flavonoids, tannins, and stilbenes, which differ in terms of the number of phenol rings they contain and the type of structural unit that connects the aromatic rings [10,11]. In the case of carob pods, the main subcategories of polyphenols are the phenolic acids (gallic acid, sinapic acid, and ferulic acid), the flavonoids (flavanols: catechin, epicatechin; flavonols: myricetin, quercetin; flavones: apigenin), and tannins (digalloyl-glucose, trigalloyl-glucose, and tannic acid). Polyphenols, as natural antioxidants, can protect cellular components from oxidative damage and, as a result, reduce the risk of a variety of degenerative diseases related to oxidative stress [12]. In particular, a reduction in cholesterol along with beneficial effects on the lipid profile of human blood have been reported in human trials after the consumption of insoluble carob dietary fiber rich in polyphenols [13]. Furthermore, the anti-cancer, anti-diabetic, and neuroprotective effects of carob are mainly attributed to its polyphenolic composition [2,14,15].

Several analytical methods have been proposed for the qualitative and quantitative determination of polyphenolic compounds in carob and its derived products. Among them, the spectrophotometric methods represent the easiest way to estimate the total phenolic content as well as the antioxidant activity of the samples. In particular, 2,2-diphenyl-1-picrylhydrazyl (DPPH), Ferric Reducing Antioxidant Power (FRAP), 2,2′-azino-bis(3-ethylbenzothiazoline-6-sulfonic acid (ABTS), and Oxygen Radical Absorbance Capacity (ORAC) assays have been previously reported in the literature to determine the antioxidant activity of carob samples [9,14,16,17,18,19]. Since antioxidant activity is directly related to phenolic content, the determination of antioxidant activity is usually followed by the determination of total phenolics, with the Folin–Ciocalteu assay being the most commonly used spectrophotometric method for the estimation of the total phenolic content in carob and its derived products [8,14,20,21,22]. The choice of two or more spectrophotometric assays is necessary for the correct evaluation of the antioxidant activity, since each test is based on a different mechanism of action and the results for the radical scavenging ability of the samples may significantly vary. The use of chromatographic and electrophoretic methods is essential for the identification and quantification of individual carob phenolic compounds. Gas chromatography (GC), capillary electrophoresis (CE), and high-performance liquid chromatography (HPLC) are the most common separation techniques applied to carob analysis [14,15,23,24,25,26]. However, due to the low volatility of phenolic compounds, GC is not preferred [10]. The most widely used technique for the identification and quantification of phenolics in carob is HPLC, coupled with either an Ultraviolet-Visible (UV) detector or mass spectrometry (MS).

Chemometrics is a useful and sophisticated tool for classifying and distinguishing samples based on certain common characteristics. Through powerful statistical modeling tools such as principal component analysis (PCA) and orthogonal partial least square-discriminant analysis (OPLS-DA) can be provide insights into classification of experimental groups based on high-dimensional measurements. Nutritional composition, volatile organic compounds, and IR spectra data have, so far, been used as indicators for the discrimination of carob-derived products as well as carob’s geographical origin [7,27,28,29,30]. Farag et al. first reported the use of secondary metabolites, as chemical markers, for the classification of carobs [31]. Then, Christou et al. achieved the chemometric classification of carob samples based on their geographical origin, using polyphenols as indicators for the chemometric analysis [6]. To the best of our knowledge, no work has, so far, been performed to classify carobs and their derived products according to their type (matrix) using chromatographic and spectrophotometric data for antioxidant activity (DPPH and FRAP) and phenolic composition as chemical markers.

Under this framework, 30 samples of carob and carob-derived products from Mediterranean countries were examined for their TPC and antioxidant capacity through DPPH and FRAP assays. In addition, the phenolic profile of the samples was investigated via HPLC analysis. The effect of the geographical origin and processing method on carob phenolic composition was also investigated. The results from TPC, DPPH, FRAP, and HPLC were used as variables in chemometrics analysis for the classification of samples based on their matrix.

## 2. Results and Discussion

### 2.1. Spectrophotometric Analysis

#### 2.1.1. Total Phenolic Content

Since polyphenolic compounds occur in carob in significant amounts and are responsible for its high antioxidant activity, the total phenolic content of carob and derived products (carob powders and syrups) was investigated. The total phenolic content (TPC) of the samples, determined using the Folin–Ciocalteu assay, is presented in Appendix A (Appendix A). TPC values ranged from 7.20 to 23.18 mg gallic acid equivalents (GAE) per gram of sample. Significant differences in TPC values were observed for each type of product studied. Specifically, the TPC ranged from 7.20–12.28 mg GAE/g, 18.19–23.18 mg GAE/g, and 9.51–14.80 mg GAE/g of sample for carob pulps, powders, and syrups, respectively, with a *p*-value < 0.001. The carob powder category presented the highest TPC, with the maximum value recorded from the roasted powder originating in Italy. It is important to highlight that roasted carob pulps and powders demonstrated higher TPC values than unroasted ones, while in the case of thermally treated (>40 °C) syrups, they presented lower TPC than non-thermally treated samples (<40 °C). The increased TPC values in roasted pulps and powders may be attributed to the breakdown of the cellular structure of plant material and the release of previously bound and insoluble phenolics during the roasting process. [4,6,32,33]. In the case of syrups, a decrease in TPC was observed during their thermal treatment, possibly due to the thermal degradation of polyphenols [6].

Data on total phenolics of carobs and their derived products vary significantly in the literature and depend on several factors such as the extraction process, geographical origin, degree of maturation, and processing method [8,20]. The TPC values obtained in the present work for carob pulp are higher than those reported by Goulas and Georgiou (2020) (351.4 ± 4.1 mg GAE/100 g carob fruit) and Gregoriou et al. (2021) (11.57 mg GAE/g extract) [14,22]. Roseiro and Duarte et al. (2013) reported a significantly higher TPC value for carob pulp using a supercritical fluid extraction (SFE) (27.1 ± 0.8 mg GAE/g dry mass) and a slightly lower value using UAE (20.4 ± 1.8 mg GAE/g dry mass) [20]. Additionally, Christou et al. (2021) (14.24 mg GAE/g carob pulp) and Kyriacou et al. (2021) (17.4 mg GAE/g dried weight) reported higher TPC values in mature carobs than those obtained in this work [8,34]. Nonetheless, the present findings on TPC are in very good agreement with several previous investigations [21,26].

Quantitative comparison of results for the TPC of carob-derived products with the literature data is even more difficult since there are not enough studies available. Furthermore, as previously mentioned, the variables affecting the obtained TPC have a decisive role and complicate the comparisons. The present findings for carob powder were found to be greater than those recorded by Petkova et al. (2017) (8.11 mg GAE/g dried weight) and in the same range as those reported by Papagianopoulos et al. (2004) (14.55–24.52 g/kg) for roasted powders [19,26]. In the case of carob syrup, our findings for the TPC appeared to be higher than those reported by Papagianopoulos et al. (2004) (3.94 g/kg) [26].

#### 2.1.2. Antioxidant Activity

It is well known that antioxidant activity cannot be properly assessed with a single assay [35]. Therefore, two commonly used assays with different operating principles and mechanisms were included in the present study. Specifically, the antioxidant activity of carobs and their derived products was evaluated in terms of (i) free radical scavenging activity, with the DPPH assay, and (ii) ferric-reducing antioxidant power, with the FRAP assay. The DPPH assay outcomes were expressed as IC_50_, which is the extract concentration (mg/mL) required to scavenge 50% of DPPH free radical. In general, a lower IC_50_ value indicates the greater antioxidant capacity of the sample, while a higher IC_50_ value is indicative of the sample’s lower antioxidant activity. As far as the FRAP assay results are concerned, they were expressed as μmol Trolox equivalents (TE)/g of the sample.

The results of DPPH and FRAP assays are demonstrated in Figure 1, with IC_50_ values ranging from 98.83 to 488.47 mg extract/mL and FRAP values from 46.36 to 144.32 μmol TE/g sample. For both assays, carob powder is the category with the highest antioxidant capacity. The sample from Italy demonstrated the greatest antioxidant activity, which is in agreement with the TPC results (Figure 1). According to the DPPH assay, the unroasted carob pulp from Paphos (Cyprus) presented the lowest antioxidant potential, while, according to the FRAP test, the thermally treated syrup from Crete (Greece) was the one with the lowest antioxidant capacity. It should be emphasized that the antioxidant activity evaluated using different antioxidant assays may vary, due to different mechanisms and operating principles of the methods [36,37]. As observed, thermal treatment significantly affects the phenolic content and antioxidant activity of the samples. In particular, non-thermal treated syrups (DPPH IC_50_ = 232.21 ± 19.58, FRAP 82.23 ± 15.81 μM TE/g product), roasted carob pulps (DPPH IC_50_ = 361.00 ± 11.92, FRAP 89.22 ± 4.71 μM TE/g product), and powders (DPPH IC_50_ = 98.83 ± 3.20, FRAP 144.32 ± 7.16 μM TE/g product) demonstrated higher antioxidant activity than thermal-treated syrups (DPPH IC_50_ = 388.05 ± 1.75, FRAP 46.36 ± 6.48 μM TE/g product) and unroasted products (DPPH IC_50_ = 488.47 ± 34.67, FRAP 73.94 ± 3.84 μM TE/g product).

Comparing the DPPH and FRAP results obtained in the present study with the literature data was a difficult task. As previously mentioned, many different parameters can affect the antioxidant capacity of the samples. For comparison purposes, the antioxidant capacity results should be expressed in terms of the same unit of measurement. DPPH values can also be expressed as % inhibition, ascorbic acid equivalents (AAE), or Trolox equivalents (TE), while FRAP values can also be described as FeSO_4_ or ascorbic acid equivalents [14,20,21,22,38,39]. The samples studied in the present work had lower antioxidant potential, in terms of DPPH radical scavenging activity, than the findings of Goulas and Georgiou (2020) (IC_50_ 1.4–2.6 mg/mL) and Papagianopoulos et al. (2004) (IC_50_ kibbles: 33.26 g/L, syrup: 26.33 g/L, roasted flours: 7.04–9.96 g/L) [22,26]. In the case of the results obtained from the FRAP assay, comparison with the other literature studies was considered impossible as the units of measurement of antioxidant capacity were different.

### 2.2. Chromatographic Analysis

#### 2.2.1. Method Validation

Calibration curve parameters, including linear ranges, regression equations, and coefficients of determination are listed in Table 1. In the concentration ranges examined, all analytes demonstrated good linearity, with the corresponding coefficients of determination being higher than 0.997. The limits of detection (LODs) and quantification (LOQs) varied from 0.1 to 0.9 μg/mL and 0.3 to 2.8 μg/mL, respectively. The precision of the method, in terms of repeatability (intraday) and reproducibility (interday), was evaluated by computing the relative standard deviations (RSDs) of the peak areas. As demonstrated in Table 1, the RSD values for intraday precision ranged from 0.32 to 1.04% and for interday precision from 0.92 to 2.68%.

#### 2.2.2. Polyphenolic Profile via HPLC Analysis

HPLC analysis was deemed essential for the identification and quantification of the main phenolic compounds in the carob pulp, powder, and syrup extracts. Furthermore, the chromatographic determination in correlation with the spectrophotometric data will assist to identify the individual compounds, which are responsible for the antioxidant activity of the extracts.

Table 2 demonstrates the qualitative and quantitative results obtained from the analysis of carob and derived products. Rutin, gallic acid, and catechin were the most abundant phenolic compounds in the extracts, as demonstrated in Figure 2. As observed, sample processing significantly affects the amount of extracted polyphenolic compounds. Roasted pulp and carob powders have the highest polyphenolic composition, while thermally treated syrups have the lowest amounts of polyphenols.

Rutin appeared to occur in greater concentrations in carob pulp, followed by catechin and gallic acid. Minor amounts of epicatechin, quercitrin, myricetin, and quercetin were detected in carob pulp, while caffeic and sinapic acid were absent. The roasting of carob pulp had a positive impact on their polyphenolic composition. This is in agreement with the findings of Christou et al., who also reported a greater amount of phenolics in roasted carob pulps than in raw pods [6]. During the roasting of the carob pulp, the concentrations of all polyphenols increased, except for quercetin, which decreased to non-measurable levels, possibly due to degradation as a result of thermal treatment [40]. It is worth to mention here that quercitrin demonstrated a significant increase during roasting (about 10 times higher concentration in roasted carob pulp). The increased concentrations of polyphenols detected in roasted carob pulp can be attributed to the breakdown of the cellular structure of the plant matrix during the roasting process, which results in the release of previously bound phenolics. The degradation of hydrolyzed tannins and the release of gallic acid units are other possible explanations for the increased polyphenolic content of roasted carob pulp. In regard to the effect of geographical origin on carob’s polyphenolic composition, even though Cyprus and Greece (Crete) have similar climatic conditions, the carob pulp from Cyprus demonstrated higher concentrations of gallic acid, catechin, epicatechin, and rutin. In contrast, quercitrin, myricetin, and quercetin were detected at similar concentration levels in samples from both countries.

The phenolic profile of the carob powder was similar to that of the carob pulp, the initial raw material, except for sinapic acid, which was detected and quantified in both roasted and unroasted carob powders. The higher phenolic composition of the unroasted powder compared to the starting material (carob pulp) may be attributed to the technological processes used to produce the carob flour. During the processing steps, bound phenolic compounds are released due to the rupture of the cellular structure of the initial raw material.

In the case of the carob pulp, the roasted powder appeared to have a higher phenolic content than the unroasted one. According to previous studies, thermal treatment of carob powder can cause hydrolysis of gallotannins and ellagitannins, which, in turn, releases gallic and ellagic acid units [4,26,32,33]. Therefore, the increased levels of polyphenols in roasted carob powders are justified and in agreement with the literature. The geographical origin of the samples also significantly influenced their polyphenolic profile. Although all samples originated from countries in the Mediterranean Basin, they appeared to have significant differences. In particular, the detected phenolic components of the Italian roasted carob powder were in lower concentrations compared to the Cypriot and Greek samples. As far as the phenolic profile of each sample is concerned, sinapic acid, quercitrin, and myricetin were not quantified in the powder from Italy, while the quantification of quercetin was not possible in powders from Cyprus and Greece.

Gallic acid was found to be the most abundant phenolic compound in carob syrups, while minor levels of catechin and epicatechin were also quantified. Μost phenolic compounds under study were not detected in either thermally or non-thermally treated syrups. Caffeic acid was quantified only in syrups originating from Greece. Although Cypriot carob pulp and powder had a higher polyphenolic content than the corresponding Greek samples, there was no such trend in the case of syrups, probably due to the different processing methods used in these two countries. As observed, thermal treatment (>40 °C) had a negative effect on the phenolic content of carob syrups. The increase in temperature during syrup processing causes the thermal degradation of polyphenolic compounds due to decarboxylation reactions. The results of the present work are in agreement with the findings of Christou et al., who also reported a lower amount of polyphenols in thermally treated syrups than in non-thermally treated ones [6].

Overall, the obtained results via HPLC analysis concur with the spectrophotometric data for antioxidant activity and TPC of the samples. The thermal treatment and geographical origin of carob and its derived products have a significant role in their polyphenolic composition.

### 2.3. Chemometric Analysis

A low-dimensional model plane’s systematic variation was estimated in a data matrix using PCA. Thirty samples were divided into three predefined groups according to their matrix (pulp, powder, and syrup). The first three principal components (PCs) of the extracted model explained 33.7% of the total variance in the data, while the first seven components explained 41.9% of the total variability. As observed from the obtained PCA scatter plot results, the samples were adequately separated based on their matrix (Figure 3). Thermally treated samples were slightly separated from the other members of their groups. This was expected, considering their different chemical composition and antioxidant activity. Moreover, the group of carob powder demonstrated a great variation, due to the quite different phenolic composition of the Italian powder, which indicates the decisive role of the samples’ geographical origin.

Furthermore, the data were analyzed statistically in order to find the correlation between the studied variables (phenolic compounds and antioxidant activity). The variable TPC demonstrated a strong correlation with FRAP, while a lower but significant correlation was also observed with DPPH. Rutin was highly correlated with catechin and epicatechin, while quercitrin and catechin appeared to have a significant correlation. Appendix A (Appendix A) presents the correlation matrix of the model’s variables.

To discriminate the samples based on their matrix, OPLS-DA was effectively performed. As demonstrated in Figure 4, the samples were well discriminated. The extracted model was described via R^2^*X* = 0.877 (the total sum of variation in *X* that is uncorrelated to *Y*), R^2^*Y* = 0.969 (the total sum of variation in *Y* explained by the model), and Q^2^ = 0.936 (the goodness of prediction). Furthermore, the OPLS-DA model was validated using cross-validation analysis (CV-ANOVA), with a *p*-value equal to 3.29 × 10^−13^. To validate the model, the misclassification table was calculated (Appendix A, Appendix A). The obtained results demonstrate 100% correct classification of all samples, according to their matrix, with a highly satisfactory Fisher’s probability reading equal to 4.8 × 10^−13^.

## 3. Materials and Methods

### 3.1. Standards and Reagents

For HPLC analysis, acetonitrile and methanol were purchased from Sigma Aldrich (St. Louis, MO, USA), while trifluoroacetic acid (TFA) was provided by Merck (Darmstadt, Germany). The analytical standards of catechin, myricetin, and quercitrin were acquired from HWI ANALYTIK GMBH (Rülzheim, Germany), while quercetin, epicatechin, sinapic acid, caffeic acid, and gallic acid were purchased from Sigma Aldrich (St. Louis, MO, USA). Rutin was provided by PhytoLab GmbH & Co (Vestenbergsgreuth, Germany).

Reagents for spectrophotometric assays, Folin–Ciocalteu reagent, sodium carbonate (Na_2_CO_3_), 2,4,6-tris(2-pyridyl)-*s*-triazine) (TPTZ), iron(III) chloride hexahydrate (FeCl_3_·6H_2_O), and sodium acetate were purchased from Sigma Aldrich (St. Louis, MO, USA). DPPH was obtained from TCI (Tokyo, Japan).

### 3.2. Plant Material and Commercial Samples

A total of thirty samples were collected and analyzed to compare their antioxidant activity and polyphenolic profile. In particular, carob pods were collected during September (ripe pods) from Cyprus (Paphos, #1–3) and Greece (Crete, #4–6). The carob fruits were washed and crushed, the seeds were separated, and the pulp was then collected and lyophilized to remove moisture (LyoDry Compact Benchtop Freeze Dryer, Mechateck Systems, Bristol, UK). The lyophilized pulp was ground into a powder (Thermomix^®^ TM5, Vorwerk, Wuppertal, Germany) and passed through a 250-μm sieve (Endecotts, London, UK) to obtain particles of uniform size. For comparison purposes, roasted pods collected from Greece (Crete, #7–9) were also studied to evaluate the effect of the roasting treatment, which was processed in the same way as raw carobs. In addition, seven carob-derived commercial products (powders and syrups) were purchased from local supermarkets. Specifically, the following products were studied: roasted powders from Italy (#10–12), Cyprus (Limassol, #13–15), and Greece (Crete, #16–18), raw powder from Greece (Crete, #19–21), thermally treated syrup from Cyprus (Limassol, #22–24) and Greece (Crete, #25–27), and non-thermally treated syrup from Greece (Crete, #28–30).

### 3.3. Extraction Procedure

Polyphenols were extracted from carob pulp and products according to the procedure described by Christou et al. [8]. Specifically, 2 g of each sample was mixed with 50 mL of 57% (*v*/*v*) acetone. The obtained mixture was then sonicated using an ultrasonic probe system (CY-500, Optic Ivymen System^®^, Barcelona, Spain) for 14 min, in a pulsed mode (5 s:5 s). After the ultrasound treatment, the obtained mixture was centrifuged at 4400 rpm for 20 min and filtered through filter paper. A rotary evaporator (RE300, Stuart^®^, Staffordshire, UK) was used to remove the organic solvent and condense the treated material before it was lyophilized and vacuum-stored at −20 °C until further analysis. The extraction procedure was performed three times for each sample.

### 3.4. Total Phenolic Content Using the Folin–Ciocalteau Method

The TPC of the extracts was determined using the Folin–Ciocalteau colorimetric method as described by Singleton et al. [41]. Prior to spectrophotometric analysis, the lyophilized extracts were redissolved in 20 mL of 57% acetone and filtered through a 0.45-μm pore size membrane filter to remove any insoluble material. Subsequently, 2 mL of properly diluted extract (1:10 (*v*/*v*) with distilled water) were mixed with 10 mL of Folin-Ciocalteau reagent (diluted to a concentration of 10% (*v*/*v*) with distilled water). After waiting 1–5 min, 8 mL of a saturated solution of Na_2_CO_3_, 7.5% (*w*/*v*) were added to the reaction mixture and shaken vigorously. The absorbance of the samples was measured after 1 h of incubation in the dark and at room temperature at 765 nm using a UV-Vis spectrophotometer (UV-1900, Shimadzu, Tokyo, Japan). The results were expressed in terms of the mg of the gallic acid equivalents (GAE) per g of dry extract and for comparison purposes as mg GAE/g of the carob pulp or product. All analyses were performed in triplicate and the results were expressed as the mean value ± standard deviation.

### 3.5. Radical Scavenging Activity by DPPH Assay

The antioxidant capacity of the carob pulp and carob products was determined using the DPPH method, as previously described by Roseiro et al. [18]. Τhe increased DPPH radical scavenging activity is indicated by the decrease in absorbance of the DPPH solution. When DPPH reacts with antioxidant components, which can donate a hydrogen atom to the DPPH radical, discoloration of the free radical is observed, from deep violet to light-yellow. The color changes are detected at 515 nm. Prior to the UV-Vis measurements, a methanolic solution of DPPH (60 μΜ) was freshly prepared. A volume of 1950 μL of DPPH solution was mixed with various concentrations of extract (50 μL) and vortexed. The samples were kept in the dark at room temperature for 30 min and the decrease in absorbance was measured using a UV-Vis spectrophotometer (UV-1900, Shimadzu, Tokyo, Japan). The same volumes of methanol and DPPH solution were mixed to prepare the blank sample. All the analyses were performed in triplicate. The radical scavenging activity (%) was calculated as ((1 – Abs_ext_)/Abs_bl_) × 100, where Abs_ext_ is the absorbance of the tested extract solution (t = 30 min) and Abs_bl_ is the absorption of the blank sample (t = 0 min). The concentration of extract (mg/mL) required to scavenge 50% of the DPPH radical was used to determine the half-maximal inhibitory concentration (IC_50_) value.

### 3.6. Ferric Reducing Antioxidant Power (FRAP Assay)

The antioxidant potential of the extracts was determined via FRAP assay through the reduction of ferric ion (Fe^3+^) to ferrous ion (Fe^2+^), using the method described by Benzie and Strain [42]. The FRAP working solution was prepared by mixing the solutions of 300 mM acetate buffer (pH 3.6), 10 mM TPTZ in 40 mM HCl, and 20 mM FeCl_3_∙6H_2_O in the ratio of 10:1:1 and incubated at 37 °C for 30 min. The extract (150 μL) was allowed to react with the FRAP working solution (2850 μL) for 30 min in the dark. The absorbance of the blue-colored mixture was then measured at 593 nm using a UV-Vis spectrophotometer (UV-1900, Shimadzu, Tokyo, Japan). All experiments were performed in triplicate. Results were expressed as mM Trolox equivalents (TE) per gram of dry extract, using a Trolox standard curve.

### 3.7. Solid-Phase Extraction

An additional step to purify the extracts was necessary, before HPLC analysis. Solid-phase extraction (SPE) is a suitable technique for purifying extracts to remove co-extracted compounds. SPE was performed according to the method described by Christou et al. [8]. A vacuum manifold system with a 12-positions rack (Visiprep™ SPE Vacuum Manifold, SUPELCO, St. Louis, MO, USA) was utilized for all SPE experiments. Discovery DPA-6S (500 mg, SUPELCO, St. Louis, MO, USA) SPE polyamide cartridges were used for the purification of the extracts. The polyamide cartridges were preconditioned with 5 mL of methanol (MeOH) and equilibrated with 5 mL of acidified methanolic solution, 20:80 (*v*/*v*) MeOH: H_2_O, acidified to pH 2.0 with HCl. The obtained lyophilized extracts were re-dissolved in acidified methanolic solution (20 mL) and then loaded into the cartridges. Subsequently, the cartridges were washed with 5 mL of acidified water (pH 2.0) to remove co-extracted compounds. Finally, the purified extracts were obtained after eluting the bound phenolics from the cartridges with an aqueous acetone mixture (2 × 5 mL, 80% *v*/*v*). The purified extracts were then dried using a rotary evaporator and lyophilizer. The dry residues were redissolved in an appropriate volume of MeOH, filtrated through a 0.45-μm pore size membrane filter, properly diluted, and injected into the HPLC system.

### 3.8. HPLC-DAD

The phenolic profile of the carob pulp and its derived products was obtained with an HPLC system from Shimadzu (Kyoto, Japan), equipped with a photodiode array detector (PDA) (SPD-M20A), a pump (LC-10AD), an autosampler (SIL-20AHT), and a thermostat column compartment (CTO-10ASVP). Chromatographic separation conditions were performed according to the procedure described by Kumar et al. [43]. The PDA detector was set at 280 nm for the determination of all polyphenolic compounds. The Venusil XSP C18 (Radnor, PA, USA) analytical column (150 × 4.6 mm, 5 μm) was equipped with a pre-column composed of the same material and kept constant at 25 °C. Milli Q water (mobile phase A) and 0.02% (*v*/*v*) TFA in acetonitrile (mobile phase B) were used as solvents with a flow rate of 1 mL/min. A gradient elution program was used, starting at 80% A for the first five min; mobile phase A was decreased linearly to 60% at 8 min and then to 50% at 12 min. Finally, mobile phase A increased linearly to 60% until 17 min and then returned to the initial composition at 21 min, where it remained constant until 25 min. Twenty microliters of each extract were injected into the system and assayed in triplicate.

### 3.9. Data Analysis

All experimental assays were performed in triplicate. The results obtained were expressed as mean values ± standard deviation (SD). The means were compared, and statistical differences were obtained through a one-way analysis of variance (ANOVA) followed by Duncan’s multiple range test at a 95% of confidence level.

For sample differentiation, the results obtained via HPLC and spectrophotometric assays were statistically processed using SIMCA 17.0.1 software (Umetrics, Umea, Sweden). Each sample was considered as an assembly of twelve variables represented by antioxidant activity and polyphenolic content. The data matrix (Dataset 12 variables × 30 observations × 3 groups) is provided in Supporting Information, Appendix A.

For the initial overview of the data, a preliminary PCA was carried out. This procedure was mainly applied to estimate the systematic variation of the observations and to reduce the dimensionality of the data matrix. The input data were mean-centered with Unit Variate Scaling (UV) and the PCA model was extracted at a confidence level of 95%.

OPLS-DA increases the quality of the classification model by separating the systematic variation of X into two parts: one that is linearly related to Y (predictive information) and one that is unrelated to Y (orthogonal information) [44]. The extracted OPLS-DA model at a confidence level of 95% was UV-scaled and log-transformed. The effectiveness of the model was described using the goodness-of-fit R^2^ (0 ≤ R^2^ ≤ 1) and the predictive ability Q^2^ values (0 ≤ Q^2^ ≤ 1). The OPLS-DA model was validated using a cross-validation of variance (CV-ANOVA), with a *p*-value < 0.05.

## 4. Conclusions

It has over the years been demonstrated that the consumption of phenolic compounds present in foods may lower the risk of health disorders due to their antioxidant activity. In recent years, carob and its derived products have gained wide attention due to their health-promoting effects, which are mainly attributed to their phenolic compounds. In this work, the polyphenolic components of carob samples (carob pulps, powders, syrups) were analyzed via HPLC, the TPC via the Folin–Ciocalteu assay, and their antioxidant capacity via DPPH and FRAP tests. The thermal treatment of the samples significantly affected their phenolic composition. Particularly, the roasting of carob pulp and powder increased the concentrations of the polyphenolic compounds, while in thermally treated syrups, the temperature had a negative impact on the polyphenolic composition. The TPC, antioxidant activity, and phenolic concentrations were utilized as chemical markers in chemometric analysis. Both PCA and OPLS-DA were performed, with 100% successful classification of samples according to their matrix in OPLS-DA. As a future perspective, the use of high-resolution mass spectrometry will reveal more polyphenolic compounds that will therefore provide more informative datasets used in chemometric analysis.

## Figures and Tables

**Figure 1 molecules-28-02269-f001:**
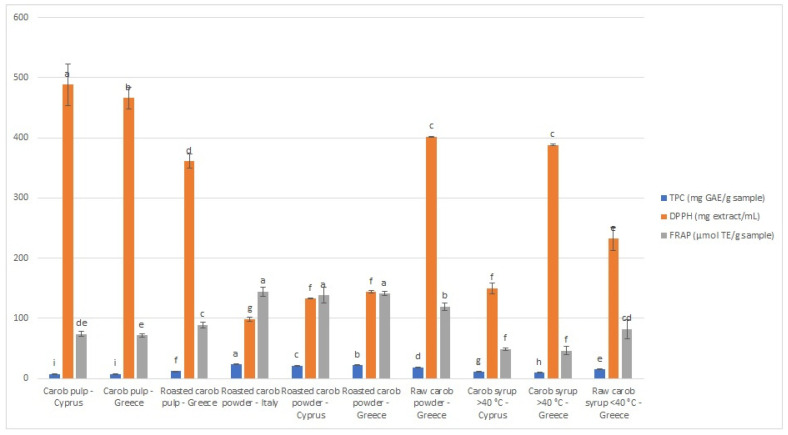
Graphical representation of antioxidant activity and total phenolic content of carob pulps and carob-derived products. (#1: unroasted carob pulp, Cyprus; #2: unroasted carob pulp, Greece; #3: roasted carob pulp, Greece; #4: roasted powder, Italy, #5: roasted powder, Cyprus; #6: roasted powder, Greece; #7: unroasted powder, Greece, #8: thermally treated syrup, Cyprus; #9: thermally treated syrup, Greece; #10: non-thermally treated syrup, Greece). Different lower-case letters indicate significant differences (*p* < 0.05) according to Duncan’s multiple range test.

**Figure 2 molecules-28-02269-f002:**
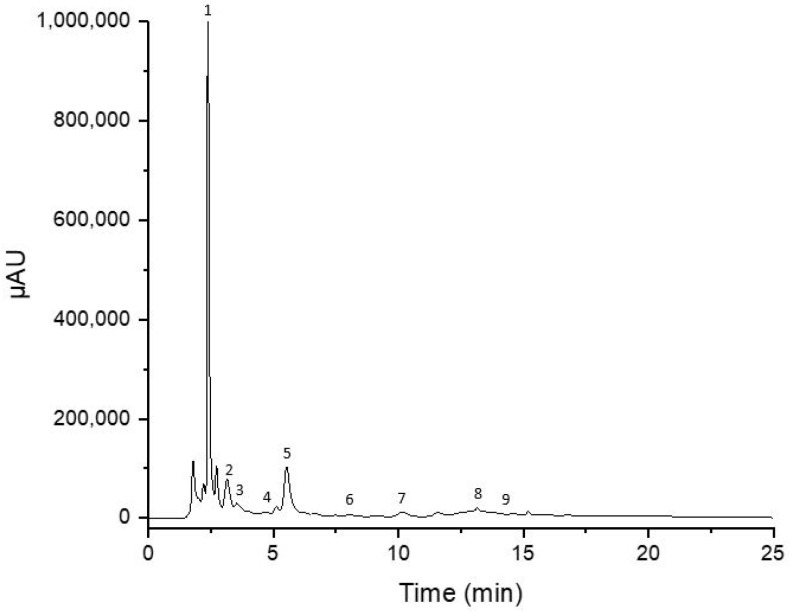
HPLC chromatogram of carob powder. (1) Gallic acid, (2) catechin, (3) epicatechin, (4) caffeic acid, (5) rutin, (6) sinapic acid, (7) quercitrin, (8) myricetin, and (9) quercetin.

**Figure 3 molecules-28-02269-f003:**
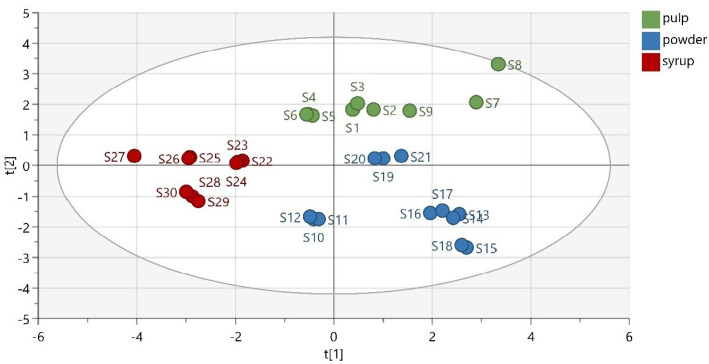
PCA scatter plot (R^2^*X* (cum) = 0.730, Q^2^ (cum) = 0.337).

**Figure 4 molecules-28-02269-f004:**
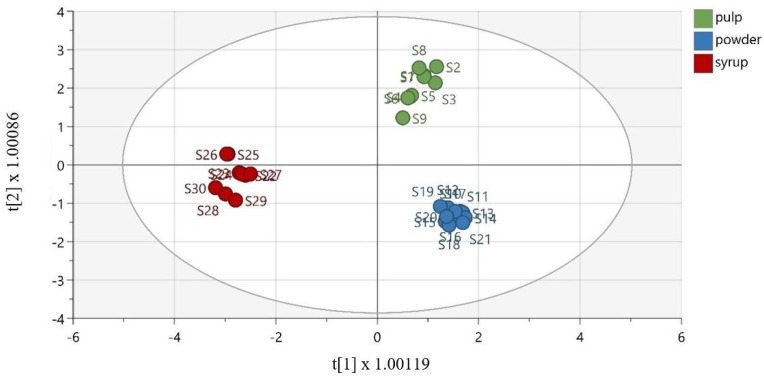
Orthogonal partial least square-discriminant analysis model (R^2^*X* (cum) = 0.946, R^2^*Y* (cum) = 0.985, Q^2^ (cum) = 0.975), discriminating the groups of carob pulps, powders, and syrups.

**Table 1 molecules-28-02269-t001:** LODs and LOQs, regression equations, coefficients of determination, retention times, repeatability and reproducibility.

Compounds	Rt ± SD	Linear Range	Regression Equation	R^2^	LOD (μg/mL)	LOQ (μg/mL)	Precision	
							Intraday	Interday
Gallic Acid	2.52 ± 0.00	0.05–0.0001	y = 99234156x − 35103	0.999	0.7	2.1	1.00	0.92
Catechin	3.34 ± 0.02	0.05–0.0001	y = 13269933x − 2012	0.999	0.3	0.9	0.95	2.53
Epicatechin	3.83 ± 0.03	0.01–0.0001	y = 16600346x + 2404	0.998	0.3	0.8	1.04	1.47
Caffeic Acid	4.54 ± 0.05	0.01–0.0001	y = 55118151x + 8713	0.997	0.3	1.0	0.47	1.96
Rutin	5.54 ± 0.07	0.05–0.0001	y = 23176073x − 12698	0.998	0.9	2.8	0.55	1.80
Sinapic Acid	7.99 ± 0.06	0.01–0.0001	y = 39382578x + 2882	0.997	0.4	1.1	0.33	1.92
Quercitrin	11.49 ± 0.08	0.01–0.0001	y = 12295377x − 1169	0.999	0.1	0.3	0.42	1.84
Myricetin	13.12 ± 0.03	0.01–0.0001	y = 25427092x + 2679	0.998	0.3	0.9	0.90	2.68
Quercetin	14.67 ± 0.04	0.01–0.0001	y = 34063703x + 42715	0.998	0.3	0.9	0.32	1.67

**Table 2 molecules-28-02269-t002:** Phenolic composition of carob pulp and carob-derived products.

Compounds	Concentration (mg/g Product)									
	Carob Pulp			Carob Powder				Carob Syrup		
	Unroasted		Roasted	Roasted			Unroasted	>40 °C		<40 °C
	Cyprus	Greece		Italy	Cyprus	Greece		Cyprus	Greece	
Gallic Acid	0.31 ± 0.01 ^e^	0.23 ± 0.01 ^e^	0.40 ± 0.02 ^d^	0.56 ± 0.07 ^c^	0.65 ± 0.09 ^c^	0.61 ± 0.05 ^c^	0.60 ± 0.05 ^c^	0.46 ± 0.01 ^d^	0.99 ± 0.03 ^b^	1.34 ± 0.03 ^a^
Catechin	0.47 ± 0.04 ^b^	0.33 ± 0.01 ^cd^	0.66 ± 0.07 ^a^	0.24 ± 0.03 ^d^	0.49 ± 0.05 ^b^	0.45 ± 0.02 ^bc^	0.55 ± 0.08 ^ab^	0.43 ± 0.04 ^bc^	0.04 ± 0.00 ^e^	0.06 ± 0.00 ^e^
Epicatechin	0.11 ± 0.02 ^c^	0.05 ± 0.00 ^de^	0.14 ± 0.05 ^bc^	0.02 ± 0.00 ^def^	0.17 ± 0.01 ^a^	0.16 ± 0.01 ^ab^	0.05 ± 0.01 ^d^	ND ^f^	0.03 ± 0.00 ^def^	0.02 ± 0.00 ^ef^
Caffeic Acid	ND ^b^	NQ ^b^	ND ^b^	ND ^b^	NQ ^b^	NQ ^b^	NQ ^b^	ND ^b^	0.03 ± 0.01 ^a^	0.03 ± 0.00 ^a^
Rutin	0.89 ± 0.06 ^a^	0.50 ± 0.03 ^c^	0.80 ± 0.09 ^a^	0.21 ± 0.03 ^d^	0.82 ± 0.16 ^a^	0.81 ± 0.08 ^a^	0.66 ± 0.08 ^b^	ND ^e^	NQ ^e^	NQ ^e^
Sinapic Acid	ND ^d^	ND ^d^	NQ ^d^	NQ ^d^	0.02 ± 0.00 ^b^	0.02 ± 0.00 ^c^	0.02 ± 0.00 ^a^	ND ^d^	NQ ^d^	ND ^d^
Quercitrin	0.03 ± 0.01 ^c^	0.04 ± 0.00 ^c^	0.34 ± 0.02 ^a^	ND ^d^	0.05 ± 0.01 ^c^	0.04 ± 0.00 ^c^	0.12 ± 0.02 ^b^	ND ^d^	ND ^d^	ND ^d^
Myricetin	0.03 ± 0.00 ^cd^	0.03 ± 0.00 ^cd^	0.05 ± 0.01 ^a^	ND ^e^	0.05 ± 0.01 ^ab^	0.04 ± 0.01 ^bc^	0.02 ± 0.00 ^d^	ND ^e^	ND ^e^	ND ^e^
Quercetin	0.02 ± 0.00 ^c^	0.02 ± 0.00 ^b^	NQ ^d^	0.03 ± 0.00 ^a^	NQ ^d^	NQ ^d^	ND ^d^	ND ^d^	ND ^d^	ND ^d^
Total	1.86	1.20	2.40	1.06	2.25	2.13	2.03	0.79	1.09	1.44

Different lower-case letters within the same row indicate significant differences (*p* < 0.05) according to Duncan’s multiple range test. ND: non-detected; NQ: non-quantified.

## Data Availability

The data presented in this study are available upon request from the corresponding author.

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
