# Peer review of "Phenolic Profile, Antioxidant Activity, and Chemometric Classification of Carob Pulp and Products"

_molecules, 2023, doi:10.3390/molecules28052269_

Round 1

Reviewer 1 Report

The current study entitled "Phenolic profile, antioxidant activity and chemometric classification of carob pulp and carob-derived products"  describes  the possible health-promoting effects of carob products. Despite the interesting features of the manuscript, some improvements are needed before considering for the publication. My specific comments are given in the PDF file.

Author Response

- The title was changed to “Phenolic profile, antioxidant activity and chemometric classification of carob pulp and products”. The word “pulp” cannot be deleted because it is not a product. It is a part of the carob bean pod.

- More information is provided in the revised Abstract. p values were also added. Please see lines 14-25.

- A sentence was added. Please see lines 54-57.

- A sentence was added. Please see lines 52 & 53.

- A sentence on PCA and OPLS-DA was added. Please see lines 87-90.

- The last sentence in Introduction was deleted. Please see lines 106 & 107.

- p value was added. Please see line 119.

- The results demonstrated in Figure 1 are the same as the ones demonstrated in Table 1. This is why it was added in the Supplementary Materials file.

- The data was added next to each sample. Please see lines 170-175.

- To the best of our knowledge, in regard to the FRAP assay, there are no studies on comparison among different carob pod parts and/or products.

- A sentence was added in Conclusions. Please see lines 454-456.

Reviewer 2 Report

My comments:

1/ please clearly state the goal; research problem; hypothesis - which should be addressed in the conclusions

2/ please highlight the novelty of the work

3/ please add the HPLC chromatogram

Author Response

- More information was added in Conclusions. Please see lines 446-451.

- The novelty is clearly stated in lines 96-99.

- A chromatogram was added. Please see Figure 2.

Reviewer 3 Report

Dear Authors, 

The manuscript entitled "Phenolic profile, antioxidant activity and chemometric classification of carob pulp and carob-derived products" written by Georgia D. Ioannou , Ioanna K. Savva , Atalanti Christou , Ioannis J. Stavrou , Constantina P. Kapnissi-Christodoulou has been submitted to Molecules journal in Natural Products Chemistry section. The choice of the journal as well as the section is well considered. 

The authors has chosen three types of carob samples for the analysis (carob pulps, powders, and syrups) of phenolic profile using high performance liquid chromatography. . Moreover, the antioxidant capacity and total phenolic content of the samples were estimated through DPPH, FRAP, and Folin-Ciocalteu spectrophoto-metric assays. The effect of thermal treatment and geographical origin of carobs and carob-derived products on their phenolic composition was assessed. Both factors significantly affect the concentrations of secondary metabolites and, therefore, samples’ antioxidant activity. 

The manuscript is well written and interesting, however I would have some remarks listed below: 

1. Analytical methods The presented methods are sufficient, the whole experiment is done and presented meticulously, easily to reperform, although it would be interesting to incorporate LC-MS or LC-MS/MS which is more precise tool to analyse the samples both qualitatively and quantitatively. Moreover, perhaps presenting a chromatogram with the results would be a good idea. I did not find it even in the supplementary materials. 

2. Chemometric Analysis A low-dimensional model plane’s systematic variation was estimated in a data matrix using PCA. To discriminate the samples based on their matrix, OPLS-DA was effectively performed. This section looks professional, good methods were used, the results are presented in Figures 2 and 3. Maybe some more figures from supplementary materials could be presented.

My overall evaluation is to accept the manuscript in present form.

Author Response

- More experiments on polyphenolic analysis are currently ongoing in our research laboratory by the use of LC-MS.

- A chromatogram was added in this revised manuscript. Please see Figure 2.

- In the current manuscript, four figures and two tables are demonstrated.
